# The Great War of Today: Modifications of CAR-T Cells to Effectively Combat Malignancies

**DOI:** 10.3390/cancers12082030

**Published:** 2020-07-24

**Authors:** Andriy Zhylko, Magdalena Winiarska, Agnieszka Graczyk-Jarzynka

**Affiliations:** Department of Immunology, Medical University of Warsaw, 02-097 Warsaw, Poland; zhylko.andrey@gmail.com (A.Z.); magdalena.winiarska@wum.edu.pl (M.W.)

**Keywords:** chimeric antigen receptor, T cells, immunotherapy

## Abstract

Immunotherapy of cancer had its early beginnings in the times when the elements of the immune system were still poorly characterized. However, with the progress in molecular biology, it has become feasible to re-engineer T cells in order to eradicate tumour cells. The use of synthetic chimeric antigen receptors (CARs) helped to re-target and simultaneously unleash the cytotoxic potential of T cells. CAR-T therapy proved to be remarkably effective in cases of haematological malignancies, often refractory and relapsed. The success of this approach yielded two Food and Drug Administration (FDA) approvals for the first “living drug” modalities. However, CAR-T therapy is not without flaws. Apart from the side effects associated with the treatment, it became apparent that CAR introduction alters T cell biology and the possible therapeutic outcomes. Additionally, it was shown that CAR-T approaches in solid tumours do not recapitulate the success in the haemato-oncology. Therefore, in this review, we aim to discuss the recent concerns of CAR-T therapy for both haematological and solid tumours. We also summarise the general strategies that are implemented to enhance the efficacy and safety of the CAR-T regimens in blood and solid malignancies.

## 1. Introduction

After so many years of searching for a cancer cure, it is hard to believe that the most potent weapon is within our bodies. Employing and redirecting T cell cytotoxicity against haematological malignancies with the help of advanced molecular engineering has recently proven to be a “magic bullet”. Chimeric antigen receptors (CARs) allow us to control the army of potent killers among the immune cells. Modified with CARs, T cells (CAR-T) demonstrated hardly ever seen anticancer efficacy [1,2,3] and CAR-T cells are now dignified as a revolution in immunotherapy. But is it a revolution or the consequence of a long-lasting process of evolution?

The modern history of cancer immunotherapy starts in Germany where the pioneers of immunotherapy were performing their first steps. The first trials employing the immune system against neoplasm were performed by Fehleisen and Busch, who reported tumour regressions in patients intentionally infected with erysipelas [4,5]. Later, American surgeon William Coley, titled the father of immunotherapy, started treating inoperable sarcoma patients with heat-inactivated bacteria [6]. Some of his patients benefited from this extraordinary treatment method [7]. However, the cause of tumour regressions remained unknown at this time. For the time being, this fledgling and unpredictable immunotherapy had to give way to the successful development of radiotherapy and chemotherapy regimens.

Immunotherapy began to resurface as adoptive cell therapy (ACT) as soon as the immune cell subpopulations and their functions were characterised. The first approach of ACT was shown by Rosenberg et al. who proved that it is feasible to harvest tumour infiltrating lymphocytes (TILs) from resected melanomas, expand them ex vivo in the presence of IL-2 and reintroduce them into patients with metastatic melanoma with a remarkable objective regression rate [8]. Unfortunately, TILs are relatively rarely detectable in tumours. Therefore, novel solutions, achieved to a great extent with the help of genetic engineering, were necessary to retarget T cell cytotoxicity. These approaches allowed researchers to either to redesign the specificity of the native T-cell receptors (TCRs) or to develop artificial CAR receptors [9]. Now, the development of CARs has become the apogee of ACT. However, the development of CARs has come a long way since the first two studies appeared at the end of the 1980s [10,11]. These studies described the construction of a TCR fused with variable antibody fragments that enabled T cell activation in a major histocompatibility complex (MHC)-independent manner. It took almost 15 years from the first description [12] to the approval of CAR-T cells as a therapeutic modality for the treatment of relapsed B-cell acute lymphoblastic leukaemia (B-ALL) in children.

Today, CAR-T cell therapy is a fundamental option for patients with CD19-positive haematological malignancies. Two types of CD19-targeting CARs were approved by the Food and Drug Administration (FDA) for treating relapsed and refractory cases. In August 2017, tisagenlecleucel (Kymriah^TM^, Novartis, Morris Plains, NJ, USA) was approved to treat B-ALL in children and young adults. In May 2018, it was approved as a treatment for diffused large B-cell lymphoma (DLBCL) [13]. The second CAR, axicabtagene ciloleucel (Yescarta^TM^, Kite, El Segundo, CA, USA) was approved by the FDA in October 2017 for the therapy of DLBCL [14]. The long-term observation of patients treated with CD19 CAR allows us to pinpoint the drawbacks of CAR-T regimens and helps to reveal important clues of cancer tumour escape from CAR-T surveillance. It becomes clear that the composition of the CAR-T cell infusion should be optimised to increase CAR safety and persistence [15,16,17]. Furthermore, it is apparent that there is no simple translation of CAR-T cell success in haematology to solid tumours. Indeed, CAR-T treatments of solid tumours are faced with the harsh conditions of the tumour milieu or poor trafficking that disrupts effective anti-tumour activity. In this review, we aim to discuss the general strategies that are currently being tested to improve CAR therapy safety and efficacy. We will also elaborate on the current urgent issues in the case of haematological CAR-T treatments and the greatest barriers in studies testing CAR-T cells in solid tumours.

Promising clinical outcomes, along with approval by the FDA, significantly increased the enthusiasm around the ACT that contributed to the spread of CAR-T therapy around the world. As CARs become more widely applied, it is necessary to be aware of their properties, their effects on T cell functionality, and their limitations and unique side effects.

## 2. Chimeric Antigen Receptor Structure

The chimeric antigen receptor (CAR) is a synthetic transmembrane receptor that is designed to activate effector cells, such as T cells or natural killer (NK) cells, in response to the recognition of surface antigen. A typical CAR construct consists of four parts differing in their function: (i) an antigen recognising domain, (ii) a hinge, (iii) a transmembrane domain and (iv) intracellular activation domains. Despite the simplicity of the modular CAR structure, it is worth noting that even small changes in its sequence could affect the biologic functions of CAR-T cells, their efficacy and persistence [18].

### 2.1. Antigen Recognising Domain

The extracellular antigen recognition domain determines the specificity of the CAR. This domain is usually composed of variable fragments of the light and heavy chain of an antibody, joined together with a peptide linker to form a single-chain variable fragment (scFv) [19]. scFv-based CARs are the most thoroughly studied “classical” antigen recognition domain of CARs. Up to now, both of the clinically approved CARs contain the same scFv, based on the FMC63 antibody [20], that enables them to recognise the epitope within the extracellular part of transmembrane glycoprotein—CD19, the marker of normal and neoplastic B cells [21]. Binding to cognate antigen in an antibody-like manner allows antigen recognition and activation of effector cells in an MHC-independent way.

Other molecules capable of specific antigen binding are also tested as extracellular domains of CAR. It was shown that ankyrin repeat proteins (DARPins) [22], a natural cytotoxic receptor NKp30 [23], membrane-tethered IL-13 [24], proliferation-inducing ligand (APRIL) [25], Fms-like tyrosine kinase 3 ligand (FLT3L) [26], granulocyte-macrophage colony-stimulating factor (GM-CSF) [27] and programmed death-ligand 1 (PD-L1) [28] can be successfully applied as target recognition domains and have shown some encouraging results in preclinical or even clinical settings [29].

### 2.2. A Hinge and a Transmembrane Domain

Effective CAR-T cell activation can also be regulated by hinge length [30]. The hinge is an extracellular part of a CAR that links the recognition and transmembrane domains. Usually derived from CD8α, CD28, IgG1 or IgG4, the hinge shows up as a multifaceted player in CAR structure responsible for functions such as antigen access, signalling and persistence of modified T cells. It was shown that appropriate adjustments in the hinge length should be introduced depending on the epitope spatial location. Epitopes located adjacent to the target cell membrane require longer and more flexible hinges than epitopes that are more easily accessible [31].

The transmembrane (TM) domain is an integral part of the CAR that connects the extracellular and intracellular portions and anchors the receptor in the cell membrane. Usually derived from inducible co-stimulator (ICOS), CD8α, CD3ζ or CD28 proteins, the transmembrane part plays a significant role in CAR function and stability. CAR constructs containing a CD3ζ transmembrane domain were shown to enhance cytokine production upon antigen stimulation by creating heterodimers with the endogenous T-cell receptor (TCR) [32]. Later, the replacement of the CD3ζ transmembrane domain with CD8α or CD28 has improved surface CAR expression, leading to increased anticancer efficacy [23]. A recent study on the third generation of CAR-T cells (described further) failed to confirm the critical influence of the TM domain on surface CAR expression, as was revealed during the investigation of the first and second-generation CAR constructs. In this study, however, the modulation of CAR-T cell’s anticancer efficacy and persistence by different TM domains was described [33].

### 2.3. Intracellular Activation Domains

Along with the recognition domain, activation domains are the most vigorously investigated parts of the CAR structure. The first described chimeric receptors, now classified as first-generation CARs, contained only a single activation domain—CD3ζ, derived from the endogenous T cell receptor [12] (Figure 1A). First-generation CAR-T cells, by binding to a target antigen, acquire the first activation signal (the one that T cells receive upon engagement of TCR to the peptide–MHC complex) [34]. The main obstacle for this CAR generation is the insufficient persistence and activation of CAR-T cells [35].

Physiologically, T cells require the first (provided by TCR) and second signal (provided by costimulatory molecules) to achieve optimal activation [36]. The costimulatory activation signal, upon CAR binding to antigen, is provided by the incorporation of the second activation domain adjacent to CD3ζ. In comparison to the first-generation, CAR-T cells modified with the second CAR generation are characterised by enhanced IL-2 secretion and proliferation capacity upon repeatable interaction with a cognate antigen [37]. CD28 and 4-1BB (CD137) are the most frequently used coactivation domains [38]. Both of the currently approved FDA CAR-T cell therapies are based on second-generation CARs—Tisagenlecleucel (Kymriah/Novartis) on a construct containing 4-1BB [13], and axicabtagene ciloleucel (Yescarta/Kite Pharma) on a construct containing the CD28 [14] coactivation domain. Despite differences within CAR constructs, both therapeutics have shown impressive clinical outcomes.

To combine benefits from different coactivation domains, the third generation of CARs, which consists of two coactivation domains, was developed [39]. T cells modified with the third CAR generation have shown enhanced anti-tumour activity and increased persistence [33]. However, the combinatory effect of two costimulatory domains within the single CAR construct does not appear to be a simple consequence of the costimulatory domain properties. For instance, the proximity of the costimulatory domain to the intracellular part of the cell membrane may determine the dominant domain and establish functional characteristics of CAR-T cells. Intriguingly, some studies report no superiority [40] of the third CAR generation and claim even worse performance than second-generation counterparts [41,42]. A deeper understanding of coactivation domain interactions within a single CAR structure is needed to create potent third-generation CARs.

The third activation signal [43], which non-modified T cells usually gain through soluble factors and typically emerges after the first two signals, has often been neglected in the context of CAR-T activation. However, the enhanced anti-tumour efficacy of T cells redirected for universal cytokine killing (TRUCKs), also called fourth-generation CARs (described in the further part of this review), and studies reporting the essential role of cytokines for full CAR-T cells activation [44] open a new path for further investigations aiming to increase CAR efficiency.

## 3. CAR T-Cell Biology

Pharmacokinetics and pharmacodynamics are critical properties of newly discovered compounds that contribute immensely to their further fate and could make or break the drug. However, in the case of CAR-T cells, the level of complicacy is even higher due to their biological features.

The basic studies on CAR-T cell biology are complicated due to many factors contributing to their properties. The phenotype of lymphocytes, the expression level of the CAR on the cell surface and even the locus of CAR DNA integration [45] can all affect CAR-T functionality. Moreover, the variations in CAR structure alter the cytophysiology of modified cells and should be taken into account while composing CAR-T cell infusions for patients. Despite this, several studies appeared and lifted the curtain of secrecy surrounding CAR-T cell biology.

To determine the cytotoxic potential of CAR-T cells, a model of T cells co-expressing the TCR and CAR with different cognate antigen specificity was created and its cytolytic ability through each of these receptors was investigated [46]. The transduction of T cells with a CAR does not influence TCR-mediated activation or proliferation. Thus, CAR-T cells remain capable of lysing multiple tumour cells after TCR or CAR-mediated engagement to a cognate antigen. Perforin and granzyme B expression, required for CAR-mediated toxicity, remained at the same level after the cell modification. The MHC-independent manner of T cell stimulation through a CAR enables activation of both CD4+ and CD8+ lymphocytes and provides effective serial killing [47]. In addition to perforin/granzyme B-mediated toxicity, upregulation of Fas ligand (FasL) expression on CAR-T cells upon activation was shown to enhance their anticancer efficacy against bystander antigen-negative tumour cells [48].

Further studies on a mechanism of CAR-T cell interaction with a target cell revealed differences within immune synapses formed after antigen recognition by TCR and CAR [49]. Following TCR-mediated antigen recognition within the MHC complex, T cells form a well-studied “bull′s eye-like” immune synapse (IS) [50,51,52]. Highly organised and stable TCR ISs contain three concentric areas, known as supramolecular activation clusters (SMACs). The central SMAC, with the TCR–MHC and Lck clusters, and peripheral SMAC, with lymphocyte function-associated antigen-1 (LFA-1) adhesion molecules responsible for IS stabilisation [53], are surrounded by the distal SMAC, where actin accumulation takes place. The study by Davenport et al. [49] revealed that the IS formed after antigen recognition by CAR differs significantly from the classical one mediated by TCR. The CAR-mediated binding to the antigen leads to the formation of a disorganised synapse with no concentric rings and randomly distributed microclusters of LFA-1, Lck and CAR–antigen complexes. The disorganised structure of CAR ISs does however have its benefits, including faster activation and deactivation of proximal and distal signals, faster lytic granule delivery to the synaptic cleft and shorter duration of the IS. These benefits of the CAR IS result in quicker CAR-T cell detachment from dying cells and movement towards another malignant cell.

Future studies are needed to investigate the influence of other coactivation domains [54] and different scFv affinities on CAR-T cell biology. This long and bumpy road lies in front of those who would dare to study the biology of the new human-made CAR-T cells.

## 4. Side Effects of CAR-T

Currently, CAR-T cell therapy brings hope to cancer patients and challenges to scientists and physicians. The remarkably rapid response to CAR-T infusions and durable remission rate observed in patients with relapsed and refractory cancer comes inseparably with unique acute toxicity. The main challenges that remain an obstacle are cytokine-release syndrome, which can be accompanied or followed by neurotoxicity and on-target off-tumour toxicity.

### 4.1. Cytokine-Release Syndrome

Cytokine-release syndrome (CRS) [55] is the most common adverse effect of CAR-T therapy and is triggered by the recognition of cognate antigen followed by the activation of T cells and bystander immune cells. Among these activated immune cells, macrophages were shown to play a crucial role in the pathophysiology of CRS [56,57]. The excessive release of pro-inflammatory cytokines such as IL-1, IL-6, IL-15 and interferon gamma (IFN-γ) [58], cause generalised immune activation which clinically manifests as high fever, hypotension, hypoxia and even multiorgan dysfunction, which can be fatal [59,60,61]. Severe CRS more commonly affects patients with bulky disease [62,63,64]. The onset of CRS typically occurs within the first week after the CAR infusion and can be managed by the use of tocilizumab (anti-IL-6R antibody) [65]. The usage of tocilizumab in the management of CRS does not seem to interfere with the anticancer efficacy of CAR-T. This observation contributed to the beginning of an ongoing clinical trial that aims to evaluate the impact of tocilizumab prophylaxis on CAR-T efficacy (NCT02906371, https://clinicaltrials.gov/ct2/show/NCT02906371). For tocilizumab non-responders and those with high-grade CRS, corticosteroids are indicated [65], despite reported interference with CAR-T efficiency [64].

Rarely, another systemic hyperinflammatory disorder may follow the CAR-T cells infusion [65]. With similar clinical manifestation and laboratory findings to CRS, hemophagocytic lymphohistiocytosis (HLH) could also appear and requires additional therapies like etoposide or cytarabine. HLH is associated with a higher mortality rate [66,67].

### 4.2. Immune Effector Cell-Associated Neurotoxicity Syndrome (ICANS)

Immune effector cell-associated neurotoxicity syndrome (ICANS) is the second most frequently seen undesirable result of CAR-T cell activation. In mild cases, patients usually experience diminished attention, language disturbance and impaired handwriting. However, in severe cases, seizures, motor weakness, increased intracranial pressure, papilledema, and cerebral oedema can also occur. Despite the hair-raising clinical picture of a patient with ICANS, signs are generally fully reversible with proper management [55]. The pathophysiology of ICANS is not fully known. There are two main hypotheses of ICANS occurrence—cytokine diffusion to the brain [68,69] and trafficking of T cells into the central nervous system [68,70]. Neurotoxicity appearing simultaneously with CRS is typically low grade and of a shorter duration, while neurotoxicity appearing after CRS (more common) is usually more severe (grade ≥ 3) and protracted [65]. Similar to CRS, ICAN management depends on the toxicity grade, with a more significant role of corticosteroids and restriction of tocilizumab usage for patients with grade ≥ 1 ICANS with concurrent CRS.

### 4.3. Cross-Reactivity of CAR-T Cells (On-Target Off-Tumour Toxicity)

The increasing number of targeted antigens tested for future CAR-T cell therapies brings into consideration the safety of the overall treatment. The main issue today is not whether we can redirect T cell cytotoxicity, but whether we can control it. The results of antigen recognition on malignant and healthy cells are also observed for the FDA approved CD19 CAR-T cell therapies. B-cell aplasia with concomitant hypogammaglobulinemia is an example of on-target off-tumour cytotoxicity of CD19 redirected modified-T cells [71,72,73]. After clearance of cancerous cells, persistent CAR-T cells not only provide surveillance, but also contribute to prolonged B cell aplasia long after cancer cell disappearance. Thankfully, loss of B cells is a relatively low price to pay for a cancer cure and can be managed with intravenous immunoglobulin infusions [74]. However, targeting more abundant molecules might lead to profound on-target off-tumour cytotoxicity, fatal in fact, as was observed during the anti-human epidermal growth factor receptor 2 (HER2)-CAR-T clinical trial [75].

## 5. General Strategies to Enhance CAR Therapy

After the first remarkable success of CAR-T cells in the field of haemato-oncology, further spreading of this therapeutic modality faced many roadblocks, one of which was the lack of tumour-specific antigens (TSAs) [76,77,78]. Due to the shortage of TSAs, most of the clinically investigated CARs are redirected against tumour-associated antigens (TAAs) [79]. To diminish off-target effects, the specific construction of CAR plasmids is indispensable. Much effort is also put into CAR construct optimisation for safety and efficacy reasons. By optimising the length of the hinge and TM domain, Ying et al. [80] increased the safety (no neurotoxicity or CRS greater than grade 1) of CAR-T cells while maintaining the anticancer efficacy. Following antigen recognition, CAR-T cells were shown to secrete lower amounts of inflammatory cytokines and were less susceptible to activation-induced cell death (AICD) when the hinge and a transmembrane domain were derived from CD8α in comparison to CD28 [81].

In the case of CARs built with a long spacer consisting of IgG1 or IgG4, certain susceptibility to AICD was observed as a result of constant fragment (Fc) antibody region recognition by FcR-bearing cells and off-target CAR-T cell activation. The mutation or elimination of FcR-recognised regions were crucial for proper CAR-T cell activation and persistence [82,83,84].

### 5.1. CARs Improvements to Target Malignant Cells

Selecting the specific antigen as a target for CAR-T cells is a crucial step in developing an effective and safe weapon against cancer. However, there are very few antigens that are expressed selectively on malignant cells. Among factors contributing to this phenomenon are: (i) the limited spectrum of antigens exhibited extracellularly, (ii) the risk of potential “off-tumour” cytotoxicity in case of more ubiquitously expressed antigens, and (iii) the relatively small database covering cancer-specific modification of antigens, such as glycosylation [85] or splicing-dependent antigen changes [86].

Modifications in the hinge region [87], TCR-like CARs or tandem CARs are among many proposed strategies to enhance malignant cell targeting. The importance of a cancer-specific modification along with the hinge length [31] was elegantly pinpointed in a study comparing CARs targeting the MUC1 glycoprotein. Due to the underglycosylation of MUC1 in malignant cells, some of the cryptic epitopes remain uncovered and can be targeted using specific antibodies. It was shown that by choosing scFv targeting these unmasked epitopes and by adjusting hinge length with an IgD linker, certain selectivity of CAR toward malignant cells can be acquired [87] (Figure 1B).

Another strategy covers the idea of creating a “universal” CAR that would be usable in different malignancies and would possibly overcome the antigen specificity issue (Figure 1C). In this approach, redirecting CAR cytotoxicity relies on labelled antibodies or small fluorescein-based adapters that recognise various tumour-associated epitopes. CAR-T cells are designed to recognise the tag or fluorescein and, therefore, their cytotoxicity is limited only to cells coated with the antibodies/adapters [88,89,90,91,92]. This approach enables not only the use of the same CAR in various malignancies, but also the targeting of many tumour antigens simultaneously. Moreover, this strategy increases the probability of overcoming intratumoural and intertumoural heterogeneity [93].

In some cases, ligands are the preferred antigen recognition domain in comparison to scFvs due to their lower affinity profile and better malignant cell discrimination [24,25,26,27] (Figure 1D). It was shown to be feasible to discriminate PD-1^high^ from PD-1^low^ targets by substituting the anti-PD-1 scFv with the PD-L1 extracellular domain [28].

Antigen binding that is unrestricted to MHC presentation, the main advantage of CARs, has serious drawbacks; it restricts the pool of targeted antigens to those presented on the cell surface. To overcome this issue, TCR-like antibodies were recently created [94] and integrated as an antigen recognition domain of CAR to form the TCR-like CAR [95,96,97]. With this strategy, CARs are capable of recognising intracellular antigens presented by MHC complexes on the surface of a cell (Figure 1E). Similarly, it was shown that fusing the extracellular part of the TCR with intracellular CAR domains can transmit the MHC-restricted recognition of antigen and trigger effector cell cytotoxicity. This strategy additionally widens the pool of cells that can be used in CAR-based approaches, as shown for the NK cell-derived cell line NK-92 [98].

The heterogeneous expression pattern of the target antigen and antigen-negative relapses in long term follow-ups, caused by the selection of malignant cells resistant to CAR-T, revealed the unmet need for targeting more than one antigen at the same time. Targeting multiple TAAs simultaneously significantly improves the discrimination of cancerous cells [99] and decreases the risk of CAR-T-resistant cancer relapse [100]. For instance, two or more [101] different CARs could be expressed within the same T cell or alternatively, separately transduced T cells could be injected sequentially or simultaneously [102]. Furthermore, “tandem CARs” were invented to link two recognition domains encoded by two scFvs in a single construct [103] (Figure 1F). Indeed, tandem CARs showed enhanced anti-tumour efficacy in comparison to T cells expressing two individual CARs [104]. This result paved the way for several ongoing clinical trials (NCT03241940, https://clinicaltrials.gov/ct2/show/NCT03241940; NCT03233854, https://clinicaltrials.gov/ct2/show/NCT03233854; NCT03448393, https://clinicaltrials.gov/ct2/show/NCT03448393) for tandem CARs.

### 5.2. Strategies that Enable CARs to Discriminate between Normal and Malignant Cells

Several methods were proposed to distinguish the variation of TAA antigen expression present on both malignant and healthy tissues [105,106,107,108]. They rely, among others, on the modulation of CAR-T cell affinity toward the antigen, dividing the full activation signal into two distinct CARs, the replacement of the coactivation domains to inhibitory ones to create iCARs, or inducible expression systems.

Apart from specific antigen recognition, CAR-T cell activation also depends on the affinity of the antigen recognition domain towards the antigen. It was shown that scFv differing in their affinity could regulate to a great extent and set a threshold for antigen recognition. The proliferation and cytokine production of CAR-T cells following antigen binding could be enhanced by increasing the affinity to the epitope [30,109]. Unnecessarily high affinity may, however, lead to exhaustion or AICD [110].

The possibility of regulating the antigen recognition threshold allows researchers to redirect the cytotoxicity of CAR-T cells towards cells with a high expression level of the target antigen while sparing the ones with low expression [111]. A tumour-associated antigen is usually expressed at a significantly higher level on tumour cells compared to non-malignant tissues. By decreasing the affinity of scFv, it becomes possible to eradicate cancerous cells and spare bystander untransformed cells with low expression of the target antigen [111,112,113] (Figure 1G).

Simultaneous targeting of multiple antigens is currently used not only to enhance anticancer efficacy, but also to increase the safety of CAR-T cell therapy. Indeed, it seems feasible to modify a single T cell with two CARs differing in the antigen specificity, where the first CAR contains only the CD3ζ activation domain and another contains only the CD28 domain [114,115,116] (Figure 1H). In this setting, the cytotoxicity mediated by CAR-T cells is restricted to target cells that have both cognate antigens. However, restricting toxicity against neoplastic cells with both target antigens opens an escape window for malignancy to evade CAR-T cells by losing one of the target epitopes [114].

As an alternative strategy, it was proposed to co-transduce a single T cell with a second CAR that has inhibitory properties (iCAR) (Figure 1I). In this approach, cytotoxicity against cells expressing the antigen recognised by the iCAR, expressed on healthy tissues, is repressed by transmitting inhibitory signals derived from the intracellular domains of checkpoint proteins, such as cytotoxic lymphocyte protein 4 (CTLA-4) or program cell death 1 (PD-1) [117].

Another strategy to limit on-target off-tumour toxicity is to express an “effector CAR” responsible for T cell activation only in defined circumstances (Figure 1J). The synthetic Notch (synNotch) system has been recently created, in which the synNotch receptor redirected against the first TAA induces the expression of the “effector CAR” specific against the other TAA [118,119]. Alternatively, by using the hypoxia-inducible CAR system, Kosti et al. recently employed the pan-ErbB antigen as a target for CAR-T cells. Despite widespread expression of the target antigen, there was no off-tumour toxicity observed along with the potent anticancer efficacy within the tumour site [120]. CAR expression can also be controlled by the administration of small molecules. A tetracycline regulation system was used to modulate the expression of a CD19 CAR. While showing minimal background leakage without doxycycline (the activator of the system), this system showed similar efficacy to CD19 CAR-T cells in terms of cytokine production and proliferation rate upon engagement to cognate antigen in the presence of doxycycline [121].

As described above, some life-threatening side effects may appear following CAR-T cell infusion. To address this problem, “safe switches” for CAR-T cells were created (Figure 1K). The co-modification of CAR-T cells with surface antigens that could be targeted with therapeutics, like rituximab and cetuximab, enables the selective clearance of modified T cells [122,123]. However, CAR-related toxicity could have fulminant manifestation and may require rapid CAR-T cell ablation, for which antibody-mediated clearance would be too slow. A far more potent and clinically applicable strategy is the co-modification of CAR-T cells with inducible caspase 9 (iCasp9) which, in the presence of a small inert molecule, dimerises and induces rapid apoptosis of T cells [124,125].

A different and ambitious strategy to overcome on-target off-tumour toxicity was recently described. Targeting CD33, the molecule expressed on healthy and neoplastic myeloid cells [126], to fight the acute myeloid leukaemia (AML) may result in off-tumour toxicity and destruction of healthy myeloid cells. Kim et al. [127] “made” CD33 specific for AML by depleting this molecule from normal haematopoietic stem and progenitor cells (HSPC) prior to autologous HSPC transplantation. After the effective engraftment of CD33-deficient HSPCs, healthy myeloid cells were resistant to CD33-CAR-T cells, enabling sufficient AML clearance without myelotoxicity.

In addition to these general issues aiming to increase CAR-T therapy safety and efficacy, some further problems, described below, were revealed in a long-term follow-up of patients receiving the approved CAR-T cell treatments.

## 6. Remaining Issues in Haemato-Oncology

Long-term observation of patients in complete response after CAR-T cell therapy revealed a depressingly high rate of disease recurrence [128]. The majority of relapses are thought to be caused by either one or all of the following reasons:

### 6.1. Insufficient CAR-T Cell Persistence and Proliferation

Adequate proliferation rate and sufficient persistence of CAR-modified T cells were shown to correlate with the durability of remission [129]. Selecting the optimal subpopulation of lymphocyte ratio [130] along with the structural CAR modifications [131,132,133] that aim to increase the viability of CAR-T cells showed encouraging results in preclinical and clinical settings [15,16,134].

Understanding the impact of costimulatory domains on a CAR-T cell phenotype opens the possibility of modifying the CAR structure to enhance the persistence of T cells [135]. Indeed, the influence of domains became evident when comparing phenotypic and metabolic changes occurring in CAR-T cells [135]. T cells that contained a CD28 domain in their CARs acquired an effector memory phenotype with brisk proliferation and glucose metabolism, which led to faster cancer clearance at the cost of limited persistence. In contrast, 4-1BB-based CAR-T cells showed slower anticancer activity with T cells differentiating predominantly into central memory cells with enhanced endurance, oxidative metabolism and increased mitochondrial biogenesis (Figure 2A1).

Clonal expansion of CAR-T cells has also been reported as a result of the lentiviral vector-mediated unintended insertion of the CAR gene into the *TET2* gene. This observation was described in a patient with a mutation in the second TET2 allele [136].

Moreover, the persistence and activity of different subpopulations of lymphocytes seem to rely on different coactivation domains. Cytotoxic (CD8+) CAR-T cell persistence was shown to depend on 4-1BB signalling, while helper (CD4+) CAR-T cells require ICOS signalling. The redirection of T cells with CAR molecules adjusted for subpopulations led to enhanced persistence and anticancer efficacy of CAR-T cells in mouse models [33] (Figure 2A2).

Preclinical investigations revealed that CAR-modified T cells with less differentiated phenotypes, like naïve or central memory, have higher anticancer efficacy [130]. By reducing the duration of ex vivo expansion of CAR-T cells, Ghassemi et al. showed enhanced anti-tumour efficacy of the modified T cells, which was caused by the less differentiated phenotype and enhanced effector functions in a murine xenograft model of ALL [137]. Additionally, the subpopulation composition of CAR-T cells emerged as a way to impact therapy outcome [17]. The first CAR-T cell therapy with a defined CD4/CD8 ratio [15,16] appeared to be applicable even in patients with severe leukopenia and is currently under the FDA approval process.

However, without potent T cells with high proliferation potential, even the perfect chimeric antigen receptor performs weakly. Preclinical experiments are often based on healthy donors′ T cells and do not take into count changes occurring during tumourigenesis. Studies indicate that during tumourigenesis, T cells acquire an exhaustion phenotype [138], characterised by a decreased proliferation capacity [139], and this change seems to be irreversible in the advanced stages of cancer. Exhausted central memory T cells have a distinct transcriptional status compared to healthy ones [140,141]. This knowledge should stimulate further studies on using healthy donor cells as a base for off-the-shelf therapeutics.

### 6.2. Relapse of Antigen-Negative Disease

The data collected during clinical trials demonstrate that CD19 antigen loss is responsible for the majority of relapses in B-ALL patients following CD19 CAR-T therapy. CD19 antigen loss was also shown to occur in NHL patients [142]. Two main mechanisms accountable for antigen loss were recently described: antigen escape and lineage switch [143].

The recurrence of phenotypically identical disease with the lack of cognate epitope characterises antigen escape (Figure 2B). There are several splice variants of CD19 described in B-ALL. Some variants lack the epitope recognised by CAR-T cells in the extracellular portion of the antigen and others lack the transmembrane region, causing the loss of CD19 surface expression [144]. CD19 splice variants in tumour cells can already be detected in patients before the CAR-T infusion [145]. CAR-T cells simply stimulate the selection of malignant cell variants resistant to therapy. However, other mechanisms of antigen escape were also reported. Braig et al. have shown that post-transcriptional alteration of CD81, a protein that regulates CD19 maturation and trafficking, leads to the loss of CD19 expression and relapse of disease [146]. On the other hand, the lineage switch mechanism depends on changes of a cancerous cell from a lymphoid to myeloid phenotype in response to the therapy [147]. The main approach to overcome these obstacles is described above and relies on the simultaneous targeting of multiple epitopes.

The most alarming issue with the lack of recognition of CD19 antigen by CAR-T cells is the semi-controllable introduction of CAR genes [148]. Unintentional transduction of a single neoplastic B cell during the production process of CAR-T led to the relapse of leukaemia with the epitope masked by the CAR on the surface of malignant cells [149] (Figure 2C). This finding illustrates the need for further improvement of manufacturing technologies to clean out engineered T cells from residual tumour cells.

### 6.3. Low Antigen Density

Low antigen density is most commonly associated with solid tumours, where the antigen expression level is very heterogeneous. Less frequently, a low antigen expression pattern is described as a problem in haematological malignancies. It was, however, portrayed by the clinical trial of CD22 CAR-T cells where some of the patients, after achieving a complete response, relapsed with malignant cells expressing low levels of CD22. At the same time, CD22 CAR-T cells were still detectable in their blood [150] (Figure 2D). This data is in agreement with previous studies reporting incomplete CAR-T cell activation after contacting the cells expressing an insufficient level of target antigen [151,152,153].

Trogocytosis is another mechanism responsible for low surface antigen level on malignant B cells [154]. CAR-T cells, after binding to a cognate antigen, may extract the antigen from a cancerous cell, causing a reversible decline of surface antigen expression level and increased resistance of malignant cells to CAR-T. At the same time, T cells with trogocytosed antigen acquire not only an exhausted phenotype, but also undergo fratricide T-cell killing caused by CD19 surface expression (Figure 2E).

## 7. Driving CARs through Solid Tumour Roadblocks

The development of CARs has undisputedly revolutionised the field of cancer therapy. However, the stunning efficacy of CAR-T cell therapy in haematological diseases has not, up until now, translated to the efficient treatment of the solid tumours, which account for approximately 90% of total cancer-caused deaths [155]. Complex cell interactions, along with the specific microenvironment within the solid mass, create a challenging labyrinth with no way-out [156]. The most significant challenges for CAR-T therapy include: (i) the lack of TSA, as described above, (ii) poor or limited trafficking of CAR-T cells to the tumour mass, and (iii) the immunosuppressive tumour microenvironment (TME).

### 7.1. Trafficking and Infiltration

Inappropriate trafficking seems to be the most potent barrier for CAR-T cells to infiltrate the tumour site. To overcome this obstacle, the direct infusion of CAR-T cells to solid masses was applied for tumours originating from different tissues [29,157,158,159,160]. Studying an orthotopic model that mimics human pleural malignancy, Adusumilli et al. showed that intrapleurally-administered CAR-T cells outperform intravenously infused cells in tumour eradication efficacy and also promote the elimination of extrapleural tumours [161]. These encouraging results contributed to the development of the ongoing clinical trial of mesothelin-targeted CAR-T cells administrated intrapleurally (NCT02414269, https://clinicaltrials.gov/ct2/show/NCT02414269). In another approach, the co-modification of CAR-T cells with a chemokine receptor that guides the CAR-T cells to the tumour site was proposed. Preclinical studies of CAR-T cells with an increased expression level of a chemokine receptor showed enhanced anti-tumour efficiency, as a result of improved migration and infiltration capacity. Forced expression of CC-chemokine receptor 4 (CCR4) on CD30-CAR-T cells enhanced the migration and anti-tumour activity in a mouse xenograft Hodgkin lymphoma model [162]. In another study, CXCR2-expressing anti-glypican-3 (GPC3)-CAR-T cells showed improved migration and activity in a xenograft hepatocellular carcinoma (HCC) tumour model [163]. Increased infiltration and anti-tumour efficacy was also shown in xenograft models of neuroblastoma and mesothelioma by co-expressing CCR2b with CARs directed against GD2 [164] and mesothelin antigens [165] (Figure 3A).

Once a T cell reaches the destination point (the tumour site), the next challenge is to penetrate the tumour extracellular matrix (ECM), which is composed of various fibrous proteins, glycoproteins, and proteoglycans. By developing CAR-T cells against fibroblast activation protein (FAP; a proteinase involved in extracellular matrix remodelling, expressed abundantly in epithelial cancers [166]), Wang et al. [167] demonstrated that clearance of cancer-associated stromal cells (CASCs) by FAP CAR-T cells inhibits tumour growth and increases tumour infiltration by host immune cells. Another study by Caruana et al. [168] showed that CAR-T cells lose the expression of heparanase (HSPE), the enzyme responsible for ECM degradation, as a result of the ex vivo expansion process. Engineered CAR-T cells with an increased expression of HSPE showed enhanced infiltration and anti-tumour efficacy (Figure 3B).

### 7.2. Inhibitory Signalling

After successfully trafficking to the tumour site, CAR-T cells are still challenged with other obstacles such as the inhibitory signals present in the TME [169]. Among many, immune checkpoints play a crucial role in cancer immune evasion. Checkpoint molecules, such as PD-1 and CTLA-4, upon ligation with their ligands, PD-L1/PD-L2 and CD80/CD86, respectively, provide inhibitory signals to T cells. Blocking the checkpoint molecules with immune checkpoint inhibitors (ICIs) appeared recently as a game-changing therapy that contributed to the revolution of immunotherapy [170]. ICIs emerge to have a striking effect in patients, especially those with melanoma and non-small cell lung cancer. The anticancer effect of ICIs is caused mainly by the unleashing of a host anticancer immune response. As CAR-T cell therapy is based on T cells, it is vulnerable to immune checkpoint blockade [169]. Several groups studied the combination of ICIs and CAR-T cells and reported enhanced anticancer efficacy [171,172]. However, ICIs may not fully block immune checkpoint activity due to the capture of antibodies by tumour-associated macrophages. Therefore, another strategy based on PD-1/PD-L1 axis disruption by PD-1 gene knockout in CAR-T cells has been recently proposed and is currently being investigated in clinical settings [173].

Instead of gene disruption, the PD-1/PD-L1 axis could also be blocked by additional CAR-T cell modification with either a dominant-negative PD-1 receptor [174] that lacks an intracellular inhibitory domain, or a switch PD-1 receptor [175] with the inhibitory domain changed to the CD28 coactivation domain. Other studies describe CAR-T cells secreting anti-PD-L1 antibodies [176] or PD-1 binding scFv [177,178]. These modifications were shown to reduce PD-1-mediated CAR-T cell exhaustion and to improve their anticancer efficacy (Figure 3C).

### 7.3. Tumour Microenvironment (TME)

Decreased nutrient availability accompanied by tumour metabolites and immunosuppressive cytokine abundance creates a fearfully perfect immunocompromised TME niche. This niche can additionally be hypoxic, prone to oxidative stress, with enzymatic systems overrode by tumour malignant signalling.

As a tumour grows, it starts to build up its vascular network, which supplies the enlarging tumour mass with nutrients and oxygen. Tumour endothelial cells (EC) were shown to contribute to the immune evasion of the solid tumours. In addition to the downregulation of adherent molecules [179], EC can induce the apoptosis of immune cells by expressing FAS ligand [180]. Several strategies were developed to destroy neoplastic vessels by targeting vascular endothelial growth factor receptor-1 (VEGFR-1) [181], VEGFR-2 [182,183,184], prostate-specific membrane antigen (PSMA) [185] or tumour endothelial marker 8, also known as anthrax toxin receptor 1 (TEM8/ANTXR1) [186] molecules to starve the tumour and enhance tumour infiltration. Due to abnormal vascularisation and rapid malignant cell proliferation, some parts of the tumour mass are exposed to hypoxic conditions. While the influence of hypoxia on CAR-T cell function remains mostly unexplored, it is known that hypoxic conditions increase adenosine production within the TME [187]. In fact, adenosine (acting mainly through the adenosine A2a receptor (A2aR)), can accumulate in concentrations [188] capable of inhibiting both native [189,190] and CAR-modified T cells. The activation of CAR-T cells through either the TCR or CAR leads to a further increase of A2aR expression [191]. Targeting A2aR with either genetic or pharmacological blockade [192], which is now thoroughly investigated as an anticancer drug [193], was shown to enhance CAR-T cell efficacy, especially in combination with PD-1 inhibitors [191].

Another pathway that leads to the inhibition of T cell activation and proliferation relies on prostaglandin E2 (PGE2) signalling, which activates protein kinase A (PKA) in a cyclic AMP (cAMP)-dependent manner [194,195]. Newick et al. showed that it is feasible to modify CAR-T cells with a small regulatory subunit l anchoring disruptor (RIAD) peptide to block the association of PKA and ezrin A-kinase anchoring protein (AKAP), which is responsible for tethering PKA to membrane lipid rafts [196], and, in turn, to increase the function of CAR-T cells and improve their trafficking into the tumour site [197].

Nutrient deprivation is another crucial feature of the TME which protects solid masses from the immune response. Decreased concentrations of arginine in serum and locally within the TME was reported in diverse types of malignancies and was shown to inhibit the expansion of both non-modified T cells and CAR-T cells [198,199,200]. The insertion of argininosuccinate synthase (ASS) and ornithine transcarbamylase (OTC) enzymes, responsible for arginine synthesis, increased CAR-T cell proliferation capacity, leading to enhanced anticancer efficacy against both haematological and solid tumours [201]. Indoleamine 2,3 dioxygenase (IDO), expressed by tumour and myeloid cells within the TME, is responsible for tryptophan conversion into metabolites that inhibit T cells. It was shown that CD19 CAR-T cells were able to inhibit the growth of an IDO-negative tumour in a xenograft lymphoma model with no effect on the growth of the IDO-positive tumour [202]. Preconditioning therapy with cyclophosphamide and fludarabine before the CAR-T cell infusion was shown to decrease the expression of IDO and enhance CAR-T cell efficacy (Figure 3D).

In solid tumours, a dysregulated redox balance with high levels of reactive oxygen species (ROS) is another hallmark of a hostile TME. CAR-T cells engineered to express catalase (CAT) [203] were shown to be resistant to ROS-induced oxidative stress [204] and additionally exerted protection towards bystander non-modified immune cells.

Additionally, it was shown that the modification of CAR-T cells with dominant-negative receptors for FasL [205] and transforming growth factor β (TGF-β) [206] or knocking out the endogenous TGF-β receptor II (TGFBR2) [207], made modified T cells resistant to proapoptotic and inhibiting signals within the TME. Modifications with switch receptors consisting of an IL-4 receptor exodomain fused with an IL-7 receptor endodomain [208,209], or with an endodomain used by IL-2 and IL-15 receptors [210], enabled CAR-T cells to convert inhibitory signals into activation signals (Figure 3E).

Another strategy to support CAR-T cells in solid tumours relies on modification of the TME to create an appropriate milieu for CAR-T persistence, the anticancer response and antigen spreading, which is needed for activation of the host immune response [211]. To re-modulate the TME, armoured CARs, known as TRUCKs [212], were developed (Figure 3F). Armoured CARs are engineered to secrete cytokines such as IL-12 [213,214,215], IL-15 [216,217,218], IL-18 [219,220,221], IL-33 [222] and IL-36γ [223], which enhance not only the CAR-T cell anticancer response, but also increase activation and infiltration of the tumour mass by host immune cells. These strategies showed encouraging results in preclinical settings and are being further investigated in a clinical trial (NCT02498912, https://clinicaltrials.gov/ct2/show/NCT02498912) [224]. However, the constitutive cytokine expression may cause severe toxicity [225]. Upon activation through either the TCR or CAR, T cells upregulate expression of the IL-23 receptor and IL-23α p19 subunit. As IL-23 consists of two subunits—IL-23α p19 and IL-12β p40 [226]—to further promote the proliferation of activated T cells via autocrine IL-23 signalling, T cells were engineered to express the p40 subunit. CAR-T cells with the incorporated p40 subunit of IL-23 showed enhanced anticancer efficacy and decreased side effects compared to IL-18 or IL-15 secreting CAR-T cells [227].

Although many steps have already been applied to tear down the barriers contributing to the weak effects of CAR-T cell therapies in solid tumours, some of these issues remain unresolved. Further preclinical studies unveiling the response of CAR-T cells in the TME are needed to precisely tailor further modifications necessary to push CAR therapies forward in the race of cancer immune-evasion.

## 8. Conclusions

CAR-based therapy is a rapidly emerging immunotherapy approach. Research on CAR-T cells is continuing at a swift pace and has been very successful in patients with blood cancers. Nevertheless, there is still room for improvement, especially a better understating of how to manage the side effects of CAR-T-based therapies. The most apparent CAR therapy problem seems to be the lack of tumour-specific antigens and the necessity to target tumour-associated antigens that are also expressed on healthy tissues. Therefore, advanced approaches for the detection of malignant cells are currently being developed, such as different strategies aimed at multiplying the antigen recognition domains in a single CAR construct, redesigning the CAR structure with ligands instead of scFvs, optimising the hinge, creating universal CARs, or even targeting intracellular antigens via TCR-like antibodies or TCR-CAR chimeras. On the other hand, there is a strong trend to discriminate malignant cells from the healthy ones by adjusting the threshold for CAR-T cell activation. This has been, to a great extent, achieved by lowering the CAR′s affinity towards the antigen. When the physiological level of the TAA on healthy cells is insufficient for CAR activation, it allows the proper recognition and elimination of malignant cells only. Another strategy in sparing healthy tissues involves logic gates. This involves splitting the activation domains into two separate CARs recognising different antigens, using inhibitory or inducible CARs, or finally incorporating “off switches” that would allow us to deactivate or eliminate CAR-T cells attacking healthy cells. Over time, it seems that a too-active response of CAR-T cells does not provide the necessary solution for treatment, but instead incorporates the unnecessary risk of adverse effects.

Both haematological and solid tumour CAR-T therapies face many obstacles that were simply difficult to foresee. After the first breath-taking effects of CAR-T cell therapy in haematological malignancies, the enthusiasm was slightly toned down by the high relapse rate following complete remissions [228]. Although many steps were undertaken to prevent these relapses, it becomes clear that many more are still needed. More comprehensive knowledge is crucial to predict factors that can influence CAR-T cell fitness and persistence in patients. Apparently, antigen loss and lineage switch appear to be an important issue in resistance to CAR therapy that needs further investigation. Interestingly, CAR-T cell biology itself presents a challenge, as it was shown that antigen loss might result from trogocytosis of the target antigen mediated by CAR-T cells. In the case of solid tumours, poor trafficking to the tumour site is still challenging but can be potentially circumvented by chemokine receptor insertion. Moreover, immune checkpoints, as well as tumour modified milieu, serve as a means to switch off CAR-T cell cytotoxicity. Thus, various strategies are applied to maintain and improve the activity of CAR-T cells in the immunosuppressive TME. All in all, it seems that the combination of multiple strategies rather than incorporating only one might be the gold standard in the field of solid tumour CAR-T cell therapy.

Despite the challenges that cancer creates, and the unsolved issues of CAR-T cells themselves, this new type of cellular therapy is now saving hundreds of lives worldwide and has the potential to save many more. Along with immune checkpoint inhibitors, CAR-T cell therapy revolutionised the field of immuno-oncology and has already changed the history of cancer treatment.

## Figures and Tables

**Figure 1 cancers-12-02030-f001:**
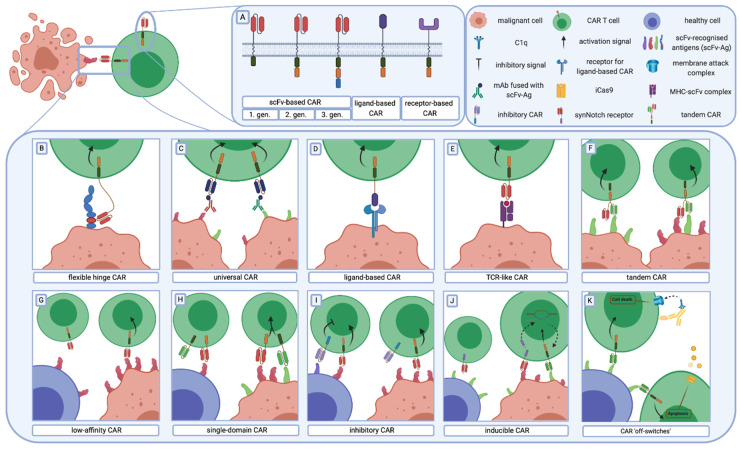
The CAR structure and general strategies to enhance CAR-T cell therapy. (**A**) Three generations of chimeric antigen receptors are based on the composition of the intracellular part. First-generation CARs contain one stimulation domain, whereas second and third generations have one or two additional costimulatory domains, respectively. Besides scFv, other molecules (ligands or receptors) are used as antigen recognition domains. (**B–F**) Strategies to increase CAR-T cell efficacy and selectivity against tumour cells. (**B**) Epitopes, located adjacent to the cell membrane, require a longer and more flexible hinge than those easily accessible for CARs. (**C**) In universal CARs, by targeting an epitope fused with an antibody, specificity relies mostly on antibody selectivity. (**D**) The use of a ligand instead of scFv as an antigen recognition domain could increase the specificity of CAR-T cells. (**E**) TCR-like CAR-T cells target intracellular cancer-specific antigens that are presented by MHC. (**F**) Tandem CARs are designed to target more than one antigen simultaneously. They recognise two different antigens and provide effective lysis of malignant cells expressing one or both cognate antigens. (**G–K**) Strategies to decrease on-target off-tumour toxicity and the probability of healthy cells lysis. (**G**) Decreased affinity toward cognate antigen enables CAR-T cells to distinguish healthy cells with a low expression level of TAAs from malignant cells with a high level of TAAs. (**H**) The division of the full activation signal to two independent CARs with different antigen specificities restrict the CAR-T cell’s cytotoxicity to malignant cells that express both targeted antigens while sparing healthy cells with only one antigen on their surface. (**I**) Additional co-modification of CAR-T cells with iCAR enables specific recognition of non-malignant cells and inhibition of toxicity against them. (**J**) The expression of the “effector” CAR is regulated by the synNotch receptor redirected against another TAA. (**K**) CAR “off-switches” are based on the co-modification of CAR-T cells with antibody-recognised molecules or with iCas9. This enables selective CAR-T cell clearance while CAR-related life-threatening side effects occur. Abbreviations: CAR—chimeric antigen receptor, scFv—single-chain variable fragment, mAb—monoclonal antibody, Ag—antigen, iCas9—inducible caspase 9, TCR—T-cell receptor, MHC—major histocompatibility complex, TAA—tumour-associated antigen, iCAR—inhibitory CAR, synNotch receptor—synthetic Notch receptor.

**Figure 2 cancers-12-02030-f002:**
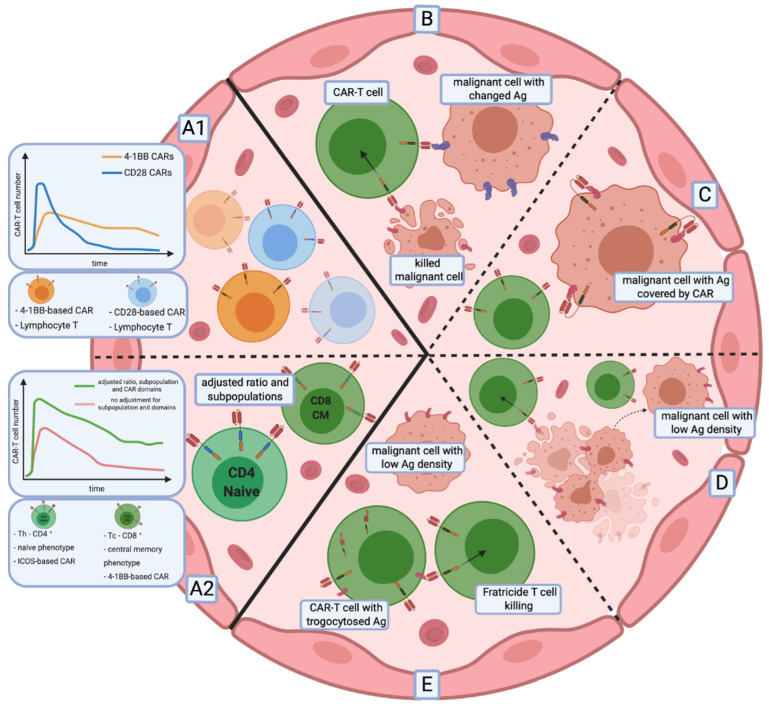
Remaining issues of CAR-T cell therapies in haemato-oncology. Relapses in a long-term observation after CAR-T cell infusion have two primary sources: CAR-T cells persistence (**A1**,**A2**) and the escape of malignant cells from immune surveillance (**B**–**E**). (**A1**) The persistence and proliferation rate of CAR-T cells depends on the co-stimulation domains. Additionally, (**A2**) subpopulation composition of the CAR-T cell infusion, as well as the phenotype of lymphocytes, influence the proliferation and persistence of modified effector cells. (**B**–**E**) Mechanisms of immune escape of neoplastic cells from CAR-T cells. (**B**) Targeted antigen changes as a result of mutation or splicing alteration. (**C**) Unintentional modification of neoplastic B cells with a CAR that masks the targeted epitope, rendering it undetectable by CAR-T cells. (**D**) Infusion of CAR-T cells triggers the selection of malignant cells with low antigen density that are resistant to effector cells. (**E**) The decrease of antigen surface level results from CAR-T cell-mediated trogocytosis, with subsequent induction of fratricide T cell killing and CAR-T cells with an exhausted phenotype. Abbreviations: Th—helper T cell, Tc—cytotoxic T cell, ICOS-based CAR—inducible co-stimulator-based CAR.

**Figure 3 cancers-12-02030-f003:**
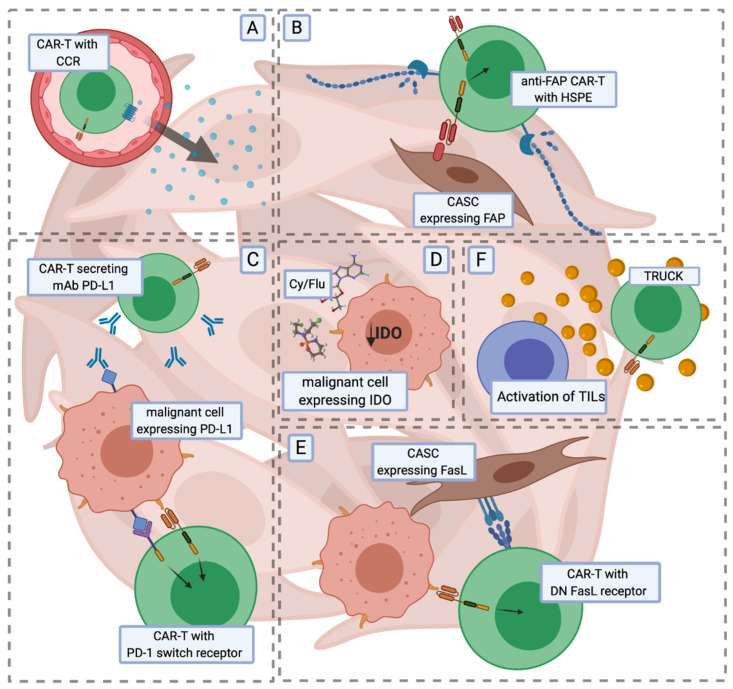
Examples of preclinical concepts supporting CAR-T cell anti-tumour activity in solid tumours. The effective eradication of solid tumours by CAR-T cells is jeopardised by impaired trafficking and infiltration into the tumour mass (**A**,**B**), competition with inhibitory signals (**C**) and reduced survival within the TME. (**A**) CAR-T cells additionally co-modified with a chemokine receptor (CCR) migrate towards and infiltrate solid masses in response to the chemokine gradient. (**B**) Forced expression of heparanase (HSPE) and redirection of CAR-T cells against fibroblast activation protein (FAP) further enhances the infiltration of solid masses. Blocking inhibitory molecules, such as PD-L1, by CAR-T cells secreting antibodies (**C**, top) or modified with switch receptors (**C**, bottom) providing activation signals leads to increased anticancer efficacy of the therapy. (**D**) Preconditional therapy with cyclophosphamide and fludarabine (Cy/Flu) decreases the expression level of IDO that inhibits T cell functions. (**E**) Modification of CAR-T cells with dominant-negative (DN) receptors for FasL leads to the resistance of CAR-T cells to proapoptotic signals present in the TME. (**F**) Armoured CARs (TRUCKs) engineered to secrete proinflammatory cytokines have enhanced anticancer efficacy and increase the activation of tumour infiltrating lymphocytes (TILs). Abbreviations: TME—tumour microenvironment, PD-L1—programmed death-ligand 1, IDO—indoleamine 2,3 dioxygenase, TRUCK—T cells redirected for universal cytokine killing, CASC—cancer-associated stromal cell, PD-1—program cell death 1, FasL—Fas ligand.

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
