# Peer review of "The Great War of Today: Modifications of CAR-T Cells to Effectively Combat Malignancies"

_cancers, 2020, doi:10.3390/cancers12082030_

Round 1
Reviewer 1 Report
The manuscript “The great war of today: CAR T cells against malignancies - immune modifications against immune evasion” reviews the current state of CAR-T therapies for haematological and solid tumours.
After providing a historic timeline of (cancer) immunotherapy, the authors in the following sections describe the molecular structure of CARs, the biology of CAR T-cells, therapy-related side effects and general strategies to enhance CAR T-cell therapy. The authors then conclude with the discussion of specific remaining challenges regarding the treatment of haematological and solid tumours.
Overall, this is a well-written review with elaborated figures that covers the important aspects of CAR T-cell – based therapies. However, it should be noted that aspects of tumor immune evasion (e.g. antigen loss) relevant for CAR-T cell therapies are only marginally covered in this manuscript. Therefore, the title of the manuscript is a bit misleading.
Moreover, some minor points concerning text editing should be addressed.
- line 149 : the acronym “TRUCKs” could already be explained here (see line 566)
- line 324 : “CARs improvements to distinguish normal cells from malignant” should be re-phrased
- line 379 : “After the effective engagement of CD33-deficient HSPCs, healthy myeloid cells were resistance to CD33 CAR-T cells”
“engagement” should be replaced with “engraftment”
Author Response
- Overall, this is a well-written review with elaborated figures that covers the important aspects of CAR T-cell – based therapies. However, it should be noted that aspects of tumor immune evasion (e.g. antigen loss) relevant for CAR-T cell therapies are only marginally covered in this manuscript. Therefore, the title of the manuscript is a bit misleading.
By tumor evasion mechanisms we understood all of the strategies displayed by tumor cells that suppress the function of CAR-T cells and subsequently render tumor cells insensitive to CAR-T therapy. We discussed briefly the clinical evidence of antigen losing in the section “Relapse of the antigen-negative disease” as well as “Low antigen density”, where the antigen is not totally lost, but its low level is beyond the CAR recognition threshold. Immune evasion can also be observed in the vicinity of tumor cells, where the concentration of different metabolic by-products, secreted enzymes or even disturbed physiological conditions such as lack of oxygen or elevated ROS levels can lead to disarming of the cytotoxic effector cells. In the revised version of the manuscript we elaborated more on the conditions present in the tumor microenvironment that efficiently block the immune response.
However, we agree with the Reviewer that the manuscript focuses more on the CAR-T cell modifications that could potentially improve the outcome of CAR-T based modalities, and therefore we decided to change the title to “The Great War of Today: Modifications of CAR-T Cells to Effectively Combat Malignancies"
Moreover, some minor points concerning text editing should be addressed.
- line 149 : the acronym “TRUCKs” could already be explained here (see line 566)
In the revised version of the manuscript we explained the acronym in line 146.
- line 324 : “CARs improvements to distinguish normal cells from malignant” should be re-phrased
We rephrased the section to Strategies that enable CARs to discriminate between normal and malignant cells in line 324 now.
- line 379 : “After the effective engagement of CD33-deficient HSPCs, healthy myeloid cells were resistance to CD33 CAR-T cells”; “engagement” should be replaced with “engraftment”
We replaced the word engagement to engraftment in line 379.
Reviewer 2 Report
The current review of CAR-T therapy is comprehensive with respect with regard to the engineering of CAR-T cells and current understanding of the ongoing work but necessarily more superficial and speculative with regards to why CAR-T has proven to have very limited utility in solid tumors thus far. The possible mechanisms responsible for failures in CAR-T therapy are highly speculative at this point and should not be presented as all inclusive or definitive as the figures appear to imply. There are many more "remaining issues" than summarized. The language of the manuscript also requires editing.
Author Response
The current review of CAR-T therapy is comprehensive with respect with regard to the engineering of CAR-T cells and current understanding of the ongoing work but necessarily more superficial and speculative with regards to why CAR-T has proven to have very limited utility in solid tumors thus far. The possible mechanisms responsible for failures in CAR-T therapy are highly speculative at this point and should not be presented as all inclusive or definitive as the figures appear to imply.
We agree with the Reviewer that the reason for the limited CAR-T cells efficacy in solid tumours is far from being properly understood. In this manuscript, we aimed to review many of the current approaches that were utilized to overcome some of the proposed mechanisms responsible for CAR-T cells low efficacy in solid tumours. The aim of our Figure supplementing the approaches in solid tumours was not meant to be closed and conclusive. We agree with the reviewer and hence decided to change the legend of the figure depicting the strategies in these tumours to Examples of preclinical concepts supporting CAR T cell antitumor activity in solid tumours. Up to now, poor CAR-T trafficking and infiltration, checkpoint inhibition of effector cells as well as the immunosuppressive function of tumour microenvironment were the most commonly raised issues influencing and limiting CAR-T efficacy in solid tumours. Therefore, in this review, we focused on presenting these issues as well as the strategies applied in preclinical settings that might enhance CAR-T cells activity.
There are many more "remaining issues" than summarized.
According to the Reviewer’s suggestion, we decided to build up the section covering the issues with CAR-T based solid tumour immunotherapies. We have thoroughly rewritten the tumour microenvironment section and added factors that were documented to inhibit CAR-T activity as well as possible solutions that might circumvent this inhibition (lines 549-613).
The language of the manuscript also requires editing.
We asked the English native speaker to revise the language of the manuscript. Accordingly, the corrections are now included in the manuscript.
Reviewer 3 Report
The authors took upon themselves a heavy task if we consider the 280 review articles on CAR-T cells this year.
Never the less I found this review very well written and easy to read and follow. There are no new insights but it is comprehensive and balanced.
Author Response
We thank the Reviewer for appreciating our work.
Round 2
Reviewer 2 Report
This review has been significantly improved. It reads well and while no review of the rapidly advancing field of CAR-T therapy can ever be entirely complete, the current manuscript presents a good summary of the field as it stands.